# Acquisition phase-specific contribution of climbing fiber transmission to cerebellum-dependent motor memory in mice

**Jewoo Seo**[1,2†], **Seung Ha Kim**[1,2†], **Jaegeon Lee**[1,2,3,4†], **Min Seok Kim**[1,2], **Yong-Seok Lee**[1,2,3,4], **Sang Jeong Kim**[1,2,3,4]*

[1]Department of Physiology, Seoul National University College of Medicine, Seoul, Republic of Korea; [2]Department of Biomedical Sciences, Seoul National University College of Medicine, Seoul, Republic of Korea; [3]Memory Network Medical Research Center, Wide River Institute of Immunology, Seoul National University College of Medicine, Seoul, Republic of Korea; [4]Neuroscience Research Institute, Seoul National University Medical Research Center, Seoul, Republic of Korea

*For correspondence:
sangjkim@snu.ac.kr

†These authors contributed equally to this work

Competing interest: The authors declare that no competing interests exist.

## eLife Assessment

This study presents potentially **valuable** insights into the role of climbing fibers in cerebellar learning. The main claim is that climbing fiber activity is necessary for optokinetic reflex adaptation, but is dispensable for its long-term consolidation. There is evidence to support the first part of this claim, though it requires a clearer demonstration of the penetrance and selectivity of the manipulation. However, support for the latter part of the claim is **incomplete** owing to methodological concerns, including the robustness of the CF marking and manipulation approach and the unclear efficacy of longer-duration climbing fiber activity suppression.

**Abstract** Climbing fiber (CF) transmission from the inferior olive (IO) triggers complex spikes (Cs) in Purkinje cells (PCs) driven by a burst of calcium spikes. In the context of motor learning, especially the compensatory optic response, CF transmission serves as an instructive signal selectively conveyed to PCs. While the significance of CF input in motor memory formation is widely acknowledged, a comprehensive understanding of its distinct contribution across different temporal windows, spanning from the initial learning phase to the retrieval period, remains incomplete. Therefore, we aimed to investigate the necessity of CF-induced instructive signals in motor learning by assessing their roles in memory acquisition, consolidation, and retrieval. We employed optogenetics to selectively inhibit CF transmission during targeted time windows. Consequently, the absence of CF-induced instructive signals during motor learning impairs memory acquisition. However, when these signals were suppressed during the consolidation and retrieval period, there was neither a loss of long-term memory nor a prevention of memory retrieval. Our results highlight that CF transmission plays a specialized and critical role primarily in memory acquisition, rather than in subsequent processes.

## Introduction

The cerebellar architecture, characterized by multiple synapses connecting diverse neuron types, enables precise adjustments within intricate neural networks (*Boyden et al., 2004*). Within this framework, the climbing fiber (CF) transmission derived from the inferior olive (IO) projecting to the cerebellar cortex is generally recognized as the primary pathway for relaying instructive signals to correct

errors and enhance motor performance, including oculomotor behaviors (*Albus, 1971*; *Frens et al., 2001*; *Ito, 1972*; *Marr, 1969*).

A prominent example of cerebellum-dependent motor learning is the optokinetic reflex (OKR), a compensatory optic response evoked by an unstable visual field (*Cahill and Nathans, 2008*; *Schweigart et al., 1997*). In OKR, adaptation to stabilize retinal images amid mismatched visual input requires the engagement of multiple neural substrates. For instance, complex spikes (Cs) in floccular Purkinje cells (PCs) driven by CF input emerge in response to unpredicted motor performance errors (*Clopath et al., 2014*; *Eccles et al., 1966*; *Goossens et al., 2004*; *Palmer et al., 2010*). Rotational visual stimulation induces Cs discharge in floccular PCs, reflecting CF-induced Cs generation in response to error signals generated from errors in oculomotor performance (*Frens et al., 2001*; *Graf et al., 1988*). Therefore, this fine-tuning of Cs in PCs is expected to regulate learning by modulating error-correcting adaptations (*Yang and Lisberger, 2014*).

Previous studies highlight a potential duality in CF involvement in cerebellar motor learning. For example, one study using a modified behavioral paradigm to suppress Cs suggested that simple spikes (Ss) alone may suffice for motor learning (*Ke et al., 2009*). In contrast, pharmacological lesioning and optogenetic inactivation of the IO impair oculomotor learning (*Pham et al., 2020*) and prevent eye blink conditioning (*Kim et al., 2020*; *Medina et al., 2002*; *Welsh and Harvey, 1998*). While CF's role in acquisition is well-supported by existing literature, its involvement in downstream memory processes like consolidation and retrieval remains contentious. This ambiguity stems from methodological limitations in previous studies, which have lacked the spatial and temporal precision to isolate CF function at distinct learning stages. Addressing this knowledge gap is essential to clarify whether CF input is uniformly critical or phase-specific in motor learning and memory.

To address this gap, we investigated the necessity of CF-induced instructive signals in motor learning by assessing their contributions across memory acquisition, consolidation, and retrieval phases. Using optogenetics, we selectively inhibited CF signals in the flocculus, a lobe-like cerebellar structure primarily responsible for ocular reflex, during targeted learning stages. Our findings provide compelling evidence for a phase-specific role of CF transmission in motor learning. This indicates that CF input is essential primarily during acquisition, with minimal involvement in consolidation and retrieval.

## Results

### Optogenetic inhibition of climbing fiber transmission derived from the inferior olive

To specifically manipulate CF signals in the cerebellar cortex, generally known to originate from the IO, we injected AAV1-CaMKIIα-eNpHR 3.0-EYFP or AAV1-CaMKIIα-EGFP into the IO of wild-type mice (*Figure 1A and B*). This approach allowed visualization of IO neuron expression and CF terminals in the brainstem nucleus and flocculus. Importantly, the selectively segregated expression in these regions enabled manipulation of CF terminals without disrupting IO somas, which, if damaged, can critically impair motor performance and ocular reflexes due to the IO's role as a prominent center of sensory integration (*Pham et al., 2020*; *Shaikh et al., 2017*; *Van Der Giessen et al., 2008*).

To confirm the effectiveness of optogenetic inhibition of climbing fiber (CF) transmission, we performed whole-cell patch-clamp recordings of Purkinje cells (PCs) in acute slices of cerebellar vermis lobules 4–5. A 593 nm yellow laser was used to activate halorhodopsin (NpHR). PCs were selected based on optical fluorescence expression (*Figure 1C and E*), and a stimulation electrode was positioned in the granule cell layer near the recorded PC. The stimulation intensity was carefully calibrated to evoke minimal CF excitatory postsynaptic currents (EPSCs). Laser illumination (3–5 mW) during stimulation resulted in a significant reduction in EPSC amplitude, indicating robust inhibition of CF transmission. Notably, this inhibition was consistent, with 9 out of 9 cells in the NpHR group exhibiting reliable suppression without failures. Importantly, laser illumination did not affect CF EPSCs in the GFP group (*Figure 1D*). Under the current-clamp mode, CF stimulation-induced spikelets were also completely abolished in the NpHR group but remained unaffected in the GFP group (*Figure 1F*). These findings establish that optogenetic inhibition effectively suppresses CF-PC synaptic transmission in vitro.

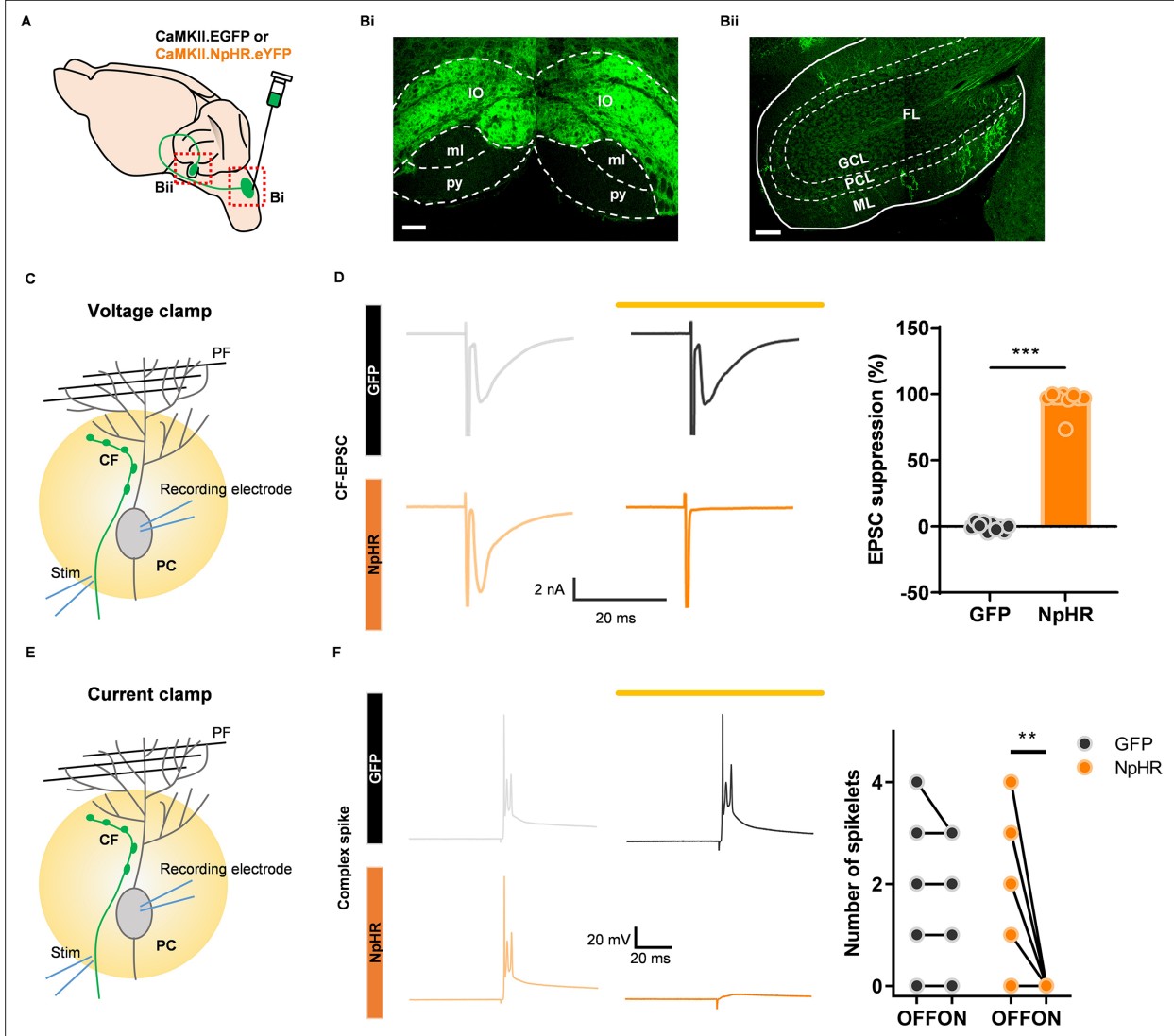

**Figure 1.** Optogenetic inhibition of climbing fiber transmission derived from the inferior olive. (**A**) Schematic diagram showing virus injection for optogenetic manipulation of CF transmission. (**B**) Bi. A virus is expressed specifically only in the IO region (IO, inferior olive; py, pyramidal tract; ml, medial lemniscus). Scale bar, 100 µm. Bii. Virus expression along CF in the cerebellar flocculus region. (GCL, granule cell layer; PCL, Purkinje cell layer; ML, molecular layer; FL, flocculus). Scale bar, 100 µm. (**C**) A scheme of optogenetic suppression using a yellow laser (593 nm, 3–5 mW) while recording CF EPSCs under voltage clamp mode. (**D**) Left, representative traces of CF EPSCs of the GFP (top) and NpHR group (bottom) with (dark line) and without (light line) opto-stimulation. Right, Quantitative analysis of suppression of CF EPSCs (GFP: n = 10 cells/2 mice, NpHR: n = 9 cells/5 mice, *** p < 0.001, Unpaired t-test). (**E**) A scheme of optogenetic suppression while recording complex spikes under current clamp mode. (**F**) Representative traces of complex spike of the GFP (top) and NpHR group (bottom) with (dark line) and without (light line) opto-stimulation. Quantitative analysis of number of complex spike spikelets (GFP: n = 8 cells/1 mouse, NpHR: n = 8 cells/5 mice, ** p = 0.002; Paired t-test). The error bars indicate ± SEM.

The online version of this article includes the following source data for figure 1:

**Source data 1.** CF-EPSC.

**Source data 2.** Complex spike.

To extend our findings in vivo, we recorded from PCs in awake, head-fixed mice during 593 nm optogenetic stimulation. AAV1-CaMKIIα-eNpHR 3.0-EYFP or AAV1-CaMKIIα-EGFP was injected into the IO followed by the creation of a cranial window above the cerebellar vermis three weeks later (*Figure 2A*). Due to technical constraints preventing direct access to the flocculus for in vivo recordings, we targeted cerebellar lobule IV/V. PCs were identified based on the detection of Cs (*Figure 2B*). Optogenetic inhibition selectively suppressed Cs firing rates in the NpHR group without affecting Ss rates, whereas no changes were observed in the GFP control group (*Figure 2C and D*). Recordings

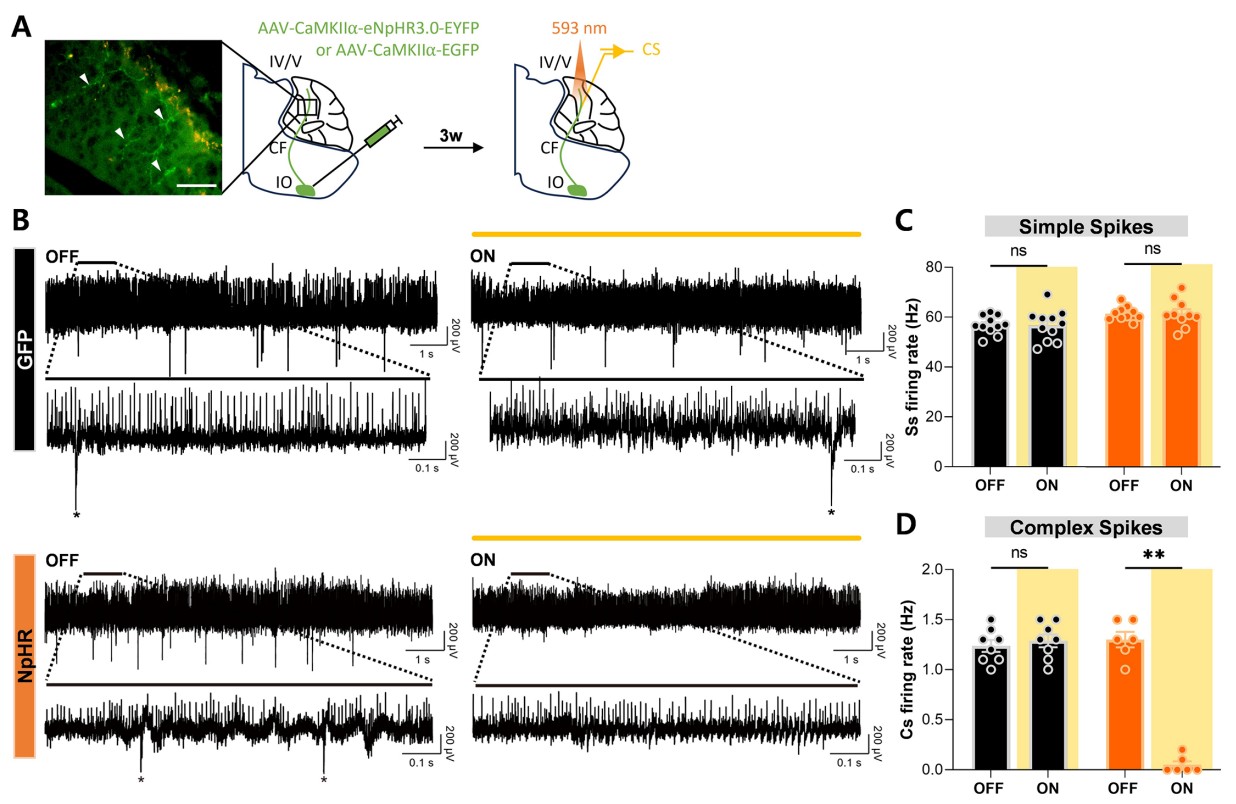

**Figure 2.** Optogenetic inhibition of climbing fiber transmission blocked Cs firing but not Ss firing in vivo. (**A**) Schematic diagram of virus injection into IO and in vivo recording during optogenetic manipulation of CF transmission. Arrow heads indicate NpHR-expressing CF terminals in the molecular layer of the cerebellar cortex. Scale bar, 50 µm. (**B**) Representative electrophysiological recording traces of PC firing from the GFP and NpHR groups, without (OFF) and with (ON) yellow (593 nm) opto-stimulation. The upper plots show 10 s traces without and with opto-stimulation, and the lower plots display a 1 s segment from the respective upper traces. Asterisks denote Cs. (**C**) Comparison of Ss firing rates during optogenetic inhibition. For the GFP group, firing rates with laser off (n = 11 cells from 4 mice) versus on (n = 11 cells from 4 mice) were not significantly different (Mann-Whitney test, p = 0.8594). Similarly, for the NpHR group, no significant difference was observed between OFF (n = 10 cells from 6 mice) and ON (n = 10 cells from 6 mice) conditions (Mann-Whitney test, p > 0.9999). (**D**) Comparison of Cs firing rates during optogenetic inhibition. In the GFP group, no significant difference was observed between OFF (n = 8 cells from 4 mice) and ON (n = 8 cells from 4 mice) conditions (Mann-Whitney test, p = 0.4563). In the NpHR group, Cs firing rates were significantly reduced during opto-stimulation (OFF, n = 6 cells from 6 mice; ON, n = 6 cells from 6 mice; Mann-Whitney test, ** p = 0.0022). Of note, 6 out of 8 recorded cells were responsive to optogenetic suppression. The error bars indicate ± SEM.

The online version of this article includes the following source data for figure 2:

**Source data 1.** IO_GFP.

**Source data 2.** IO_NpHR.

**Source data 3.** IO_GFP.

**Source data 4.** IO_NpHR.

from the same neurons during both laser-off and laser-on conditions demonstrated consistent and robust suppression of Cs activity throughout the 40-min optostimulation period. Although continuous recordings over several hours were not feasible, the stability and sustained suppression observed at 40 min strongly suggest that the manipulation would remain effective during the extended durations required for behavioral experiments. These findings confirm that optogenetic inhibition of NpHR-expressing CFs selectively suppresses Cs activity in awake mice without altering Ss activity. Furthermore, they validate the feasibility and specificity of using optogenetic interventions to block CF transmission both in vitro and in vivo, reinforcing the reliability of this approach for studying CF contributions to cerebellar function and behavior.

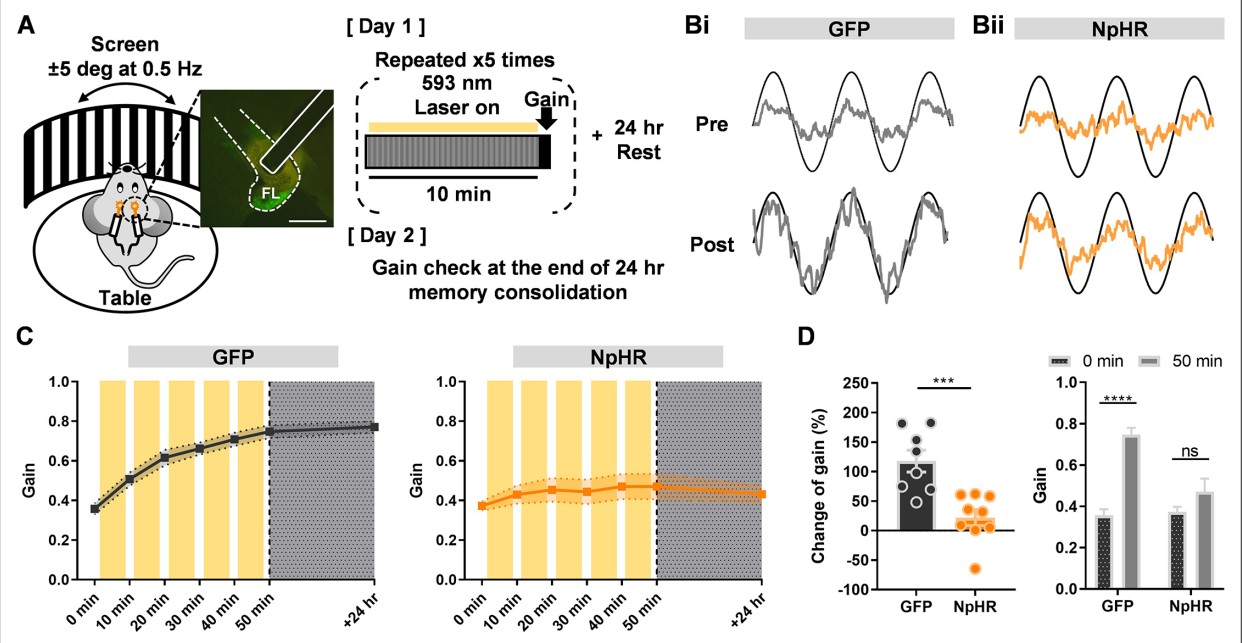

**Figure 3.** Inhibition of climbing fiber transmission during memory acquisition of OKR. (**A**) Illustration of the optokinetic reflex behavioral test (left) and experimental scheme (right). For training, visual stimulation was given by constant rotation of the screen, while the table, where a mouse's head is fixed, remained stationary. For day 1, a mouse was trained for a total of 50 minutes, and on day 2, an amount of sustained memory was validated via gain check. Scale bar, 100 μm. (**B**) Representative traces of the screen (black sinusoidal curve) and eye movements (grey sinusoidal curve) divided into the GFP (i) and NpHR (ii) groups. Top traces (pre) were acquired before learning, and bottom traces (post) were obtained after 50 min of learning. The vertical scale bar represents 10 degrees per second and the horizontal scale bar represents 0.5 s. (**C**) The learning curve from 0 to 50 min, and +24 hr period. Change of gain is indicated at each time point (+10 min increment). Yellow boxes indicate opto-stimulation (12.5 58 mW for 10 min/session, total of 5 sessions). The left graph represents the GFP (n = 8) and the right graph indicates the NpHR (n = 9) group. (**D**) Comparison of gain changes between the GFP and NpHR groups (GFP, n = 8 mice; NpHR, n = 9 mice). Percentages of gain increment from 0 to 50 min were calculated. The change of gain of the GFP group was significantly larger than that of the NpHR group (left; Unpaired t-test, p = 0.0007). The gain of the GFP group was significantly increased from 0 to 50 min (right; Paired t-test, **** p < 0.0001), while the NpHR group showed no significant improvement of gain from 0 to 50 min (right; Paired t-test, p = 0.1046). The error bars indicate ± SEM.

The online version of this article includes the following source data for figure 3:

**Source data 1.** OKR_Learning Curve_NpHR.

**Source data 2.** OKR_Learning Curve_GFP.

**Source data 3.** Change of Gain.

**Source data 4.** Pre vs Post with Opto.

## Inhibition of climbing fiber transmission during memory acquisition

An evaluation of CF's direct contribution in a region-specific manner has not been performed. Thus, after confirming the viability of optogenetic inhibition of CF, we investigated whether the elimination of CF transmission in the cerebellar flocculus affects cerebellar motor learning. When images on the retina are unstable due to mismatched visual inputs, CF signals are conventionally perceived to be responsible for sending error signals to the cerebellar cortex to improve ocular reflexes (*Bloedel and Bracha, 1998*; *Frens et al., 2001*; *Zang and De Schutter, 2019*). Using the same mouse model shown in *Figure 1*, we surgically implanted an optical fiber in flocculus to suppress CF transmission in vivo during OKR (*Figure 3A*). Head-fixed mice were placed in a rotating drum painted with a black striped pattern to provide visual input, and its rotating motion elicited an unstable retinal image. Hence, in the beginning, the trace of the mouse's eye movement represented in a sine curve was only partially overlapping with the trace of the drum's kinetics, but as the mouse underwent more sessions of OKR learning, the trace of eye movement became almost identical (*Figure 3Bi*). Notably, the group with inhibited CF transmission exhibited only slight changes or no improved trace of eye movements (*Figure 3Bii*). Overall, the results demonstrated that CF transmission is required for motor memory acquisition.

This indicates that CF transmission is essential for cerebellar motor learning by presenting two contrasting learning curves. The control group exhibited an increasing trend in the learning curve, whereas the NpHR group's learning curve was almost flat, which reflects no change of gain (*Figure 3C*). The gain value through the learning sessions was distinctly increased in the control group, whereas the NpHR group showed no significant change in gain (*Figure 3D*). Furthermore, from 0 to 50 min period of learning, the control group demonstrated almost a doubling in gain, whereas the NpHR group showed little or almost no change of gain (*Figure 3E*). These results indicate that CF transmission is required for the proper acquisition of motor memory.

### Inhibition of climbing fiber transmission during memory consolidation and retrieval phase

When memories are initially acquired, they are generally unstable; hence, it requires some time to shape a stabilized long-term memory (*Attwell et al., 2002*). Therefore, memories undergo multiple processes that allow distinct shifts from short-term to long-term memory, leading to the construction of adapted behavior (*Shutoh et al., 2006*). Having confirmed the active involvement of CF transmission during memory acquisition, we sought to understand if its role is limited to the acquisition period or extends to memory transfer during the consolidation phase for long-term storage. We implemented identical surgical and behavioral procedures as previously conducted. However, instead of optogenetically inhibiting the transmission during memory acquisition, we suppressed CF activity in two distinct optogenetic manipulation schemes: one targeting short-term inhibition (30 min) and the other for long-term inhibition (6 hr) (*Figure 4A and D*). This approach was based on findings that long-term memories associated with cerebellum-dependent behaviors are vulnerable to disruption within specific time windows of consolidation, spanning from minutes to hours (*Cooke et al., 2004*; *Titley et al., 2007*). Learning curves confirmed the successful gain enhancement, validating the integrity of memory acquisition (*Figure 4B and E*). Once the gain was robustly enhanced, optogenetic inhibition of CF transmission during consolidation did not yield any change in the maintenance of memory, irrespective of the duration of manipulation. Consequently, long-term memory was well-established in both NpHR and GFP groups (*Figure 4C and F*). In addition, we verified whether CF signaling is required during the retrieval phase (*Figure 4G*). When mice, fully trained in the optokinetic reflex, displayed increased gain, disrupting CF signaling did not affect the retrieval of motor memory (*Figure 4H*). This divergence underscores a specialized role for CF transmission in the early stages of motor learning, with subsequent phases relying on distinct neural mechanisms. The results highlight phase-specific contributions of CF-driven error signaling, aligning with the hypothesis that memory consolidation and retrieval depend on downstream circuits independent of CF activity.

## Discussion

Motor learning is a finely tuned process, yet its underlying cellular mechanisms operate with remarkable precision, adapting selectively to task demands. This adaptability relies on CF activity, originating from the inferior olive, which instructs cerebellar circuits to reduce motor errors and enhance performance. CF-driven Cs in PCs have long been associated with motor learning, but how CF signals vary across learning stages remains less understood. Our study uncovers a distinct role for CF signaling that challenges traditional views: CF input is essential for driving motor adaptation during memory acquisition but surprisingly dispensable during consolidation and retrieval. This phase-specific pattern of CF transmission in cerebellum-dependent learning indicates that its role is not uniform over time, as previously thought, but instead selectively tunes cerebellar circuits to optimize adaptation.

By inhibiting CF transmission optogenetically, we demonstrated that CF activity is crucial for enhancing motor gain in the early phases of memory acquisition, where substantial adjustments and corrections are necessary. Conversely, CF input was found to have a minor influence during memory consolidation and retrieval, indicating that different mechanisms may dominate as motor memories become stable and transition to downstream circuits. This finding refines our understanding of CF function: CFs prompt learning adjustments but do not play a key role in the later phases of memory storage or recall, thereby altering our perspective on the cerebellum's role in memory formation over time.

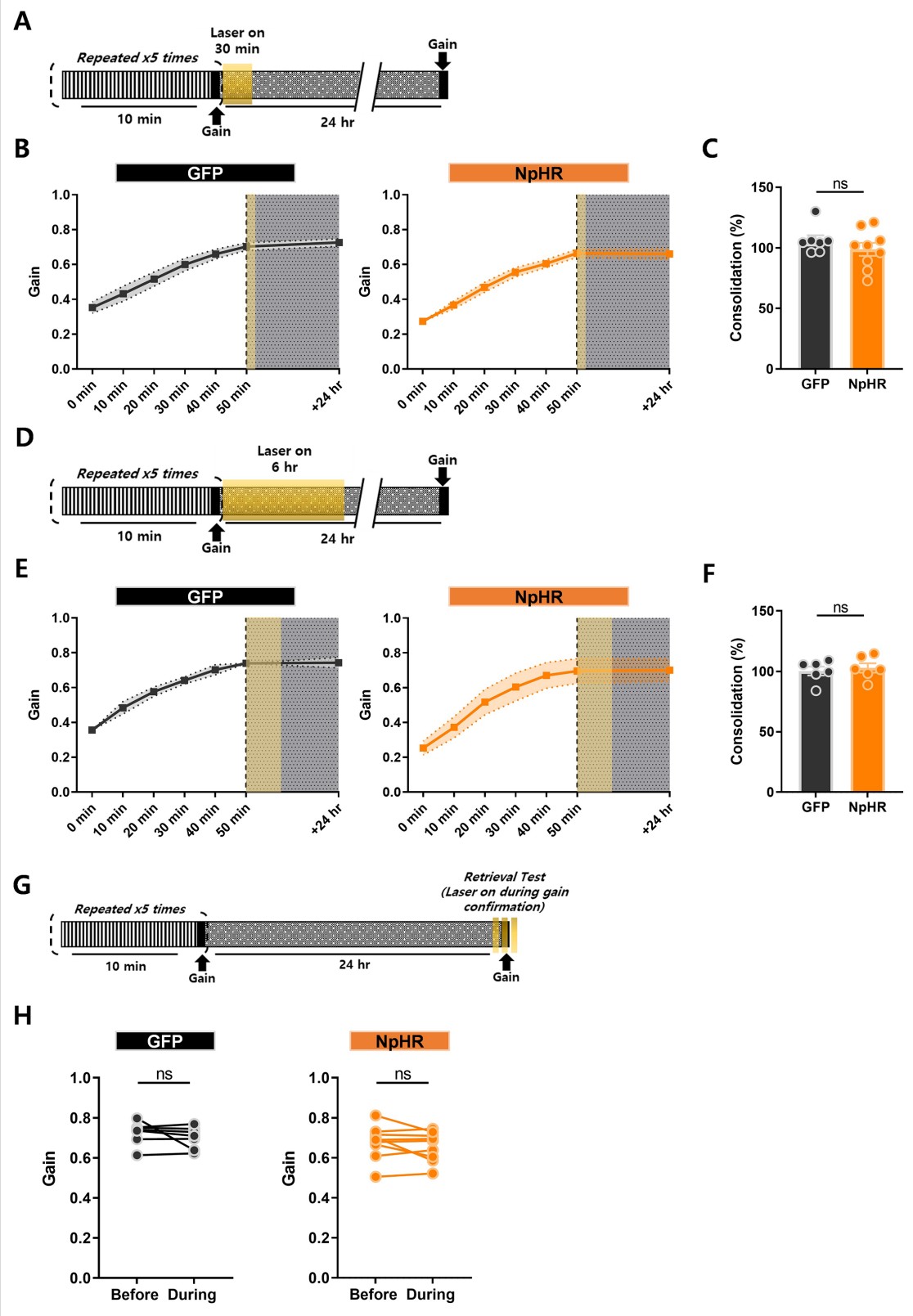

**Figure 4.** Inhibition of climbing fiber transmission during memory consolidation and retrieval phase. (**A**) Illustration of the experimental scheme. 10 min of optokinetic reflex training was repeated 5 times, and then opto-stimulation (12.5 mW) was given at 0 min post-learning for 30 min. (**B**) The learning curve from 0 to 50 min, and +24 hr period. Change of gain is indicated at each time point (+10 min increment). A yellow box indicates opto-stimulation at 0 min post-learning. The left graph represents the GFP and the right graph indicates the NpHR group. (**C**) Comparison of consolidation percentage

*Figure 4 continued on next page*

*Figure 4 continued*

between the GFP and NpHR groups (GFP group, n = 7 mice; NpHR group, n = 9 mice). The percentage of sustained memory after 24 hr was calculated. No difference was observed between the groups (p = 0.3173; Unpaired t-test). (**D**) Illustration of the experimental scheme. 10 min of optokinetic reflex training was repeated 5 times, and then opto-stimulation (12.5 mW) was given at 0 min post-learning for 6 hr. (**E**) The learning curve from 0 to 50 min, and +24 hr period. Change of gain is indicated at each time point (+10 min increment). A yellow box indicates opto-stimulation at 0 min post-learning. The left graph represents the GFP group, and the right graph indicates the NpHR group. (**F**) Comparison of gain changes between GFP and NpHR groups (GFP group, n = 6 mice; NpHR group, n = 6 mice). Percentages of gain increment from 0 to 50 min were calculated. No difference was observed between the groups (Unpaired t-test, p = 0.6405). (**G**) Illustration of the experimental scheme. Following proper motor learning on day 1, the remaining gain value was checked on day 2, after testing 1st gain retrieval without opto-stimulation, 2nd gain retrieval was tested with opto-stimulation to validate the effect of CF inhibition on gain retrieval (opto-stimulation was given 3 times for 24 s each at 12.5 mW). (**H**) Gain retrievals before and during opto-stimulation were compared. The left graph represents the GFP group, and the right shows the NpHR group (GFP group, n = 6 mice; NpHR group, n = 8 mice). There was no significant variation between the groups (NpHR group, p = 0.2439; GFP group, p = 0.3180; Paired t-test). The error bars indicate ± SEM.

The online version of this article includes the following source data for figure 4:

**Source data 1.** OKR_Learning Curve_NpHR_0 min.

**Source data 2.** OKR_Learning Curve_GFP_0 min.

**Source data 3.** Consolidation Percentage_0 min.

**Source data 4.** OKR_Learning Curve_NpHR_6 hr.

**Source data 5.** OKR_Learning Curve_GFP_0 min.

**Source data 6.** Consolidation Percentage_6 hr.

**Source data 7.** Retrieval_24 hr_NpHR.

**Source data 8.** Retrieval_24 hr_GFO.

Motor memories gained through cerebellar-dependent learning are thought to transfer to the vestibular nuclei for long-term retention (*Jang et al., 2020*; *Seo et al., 2024*; *Shutoh et al., 2006*). By employing optogenetic inhibition of CF transmission at various stages of learning, we identified the phase-specific role of CF signaling. CF activity was crucial for motor adaptation, such as gain increase training, during the acquisition phase but was not needed during consolidation and retrieval. This suggests that the primary function of CF input lies in initial memory formation rather than in subsequent memory processing.

Our results suggest that CF-dependent and CF-independent mechanisms may operate in a context-sensitive manner during cerebellar learning, consistent with established insights derived from the vestibulo-ocular reflex (VOR) and OKR systems. Specifically, CF signaling appears essential in tasks involving substantial, persistent errors requiring precise error correction. For instance, in the VOR, the increase in gain is contingent upon CF-driven complex spikes, which provide adaptive signals in response to large error inputs (*Ke et al., 2009*). Conversely, CF-independent processes can facilitate reductions in VOR gain, wherein smaller or more gradual adjustments are adequate. This selective role of CF signaling reflects a broader functional adaptation within cerebellar circuits, permitting the engagement of different pathways according to the demands of motor learning. The findings related to our OKR results correspond with this notion, emphasizing the involvement of CF-dependent pathways in tasks characterized by substantial error correction throughout the acquisition phase. In contrast, CF-independent mechanisms likely enable incremental adjustments of lesser magnitude, thereby promoting efficient motor adaptation in the absence of CF input. This delineation of responsibilities between CF-dependent and CF-independent processes highlights the cerebellum's adaptive plasticity, wherein CF input is selectively activated based on the magnitude and persistence of motor errors. The cerebellum maintains a balance between precision and flexibility by dynamically alternating between these mechanisms, enhancing motor adaptation across various behavioral contexts. This interplay offers a versatile framework for motor learning in cerebellar-dependent tasks, encompassing VOR, OKR, and likely other forms of sensorimotor adaptation.

In addition to their robust excitatory drive on PCs, CFs shape cerebellar computation by recruiting molecular-layer interneurons to impose feedforward inhibition. Optogenetic activation of molecular-layer interneurons, which silences PCs, markedly suppresses CF-evoked dendritic calcium spikes; thereby, CF transmission may dynamically gate their own instructive signals, shifting the polarity and magnitude of synaptic plasticity at PF–PC synapses (*Rowan et al., 2018*). Beyond the cerebellar

cortex, anatomical tracing reveals that the same IO neurons that innervate the flocculus also send axon collaterals to the vestibular nuclei (*Balaban et al., 1981*), suggesting that CF signals can simultaneously modulate cortical and vestibular outputs. While the behavioral consequences of this direct CF-vestibular nucleus pathway remain to be defined, it provides a parallel channel through which IO teaching signals may coordinate cortical plasticity with adaptive vestibular reflexes.

The process of systems consolidation, facilitated by the medial vestibular nuclei (MVN), a downstream circuitry of floccular PCs, is essential for long-term memory storage (*Shutoh et al., 2006*). Given that CF activity modulates floccular PC firing, we hypothesized that CF signaling might indirectly influence MVN activity during memory consolidation and retrieval. However, contrary to this expectation, our results revealed no substantial effect of CF transmission on long-term memory formation or recall, suggesting that CF signaling does not directly impact MVN function. Instead, previous studies suggest that PC intrinsic plasticity supports memory consolidation (*Seo et al., 2024*), while granule cell activity contributes to memory retrieval (*Wada et al., 2014*). These findings reinforce the notion that CF signaling primarily facilitates the acquisition phase of motor memory, while simple spiking in PCs may encode timing and frequency information crucial for stabilizing memory traces in downstream circuits. Ss could either complement or function independently of CF transmission, especially during phases when error correction demands are minimal. Mechanistically, suppressing CF signaling likely mimicked key aspects of long-term depression (LTD) at CF-PC synapses, such as reducing the amplitude of Cs (*Hansel and Linden, 2000*). Under normal conditions, CF input promotes parallel fiber-PC (PF-PC) LTD by driving calcium influx during error correction, such as in response to retinal slip (*Coesmans et al., 2004*). CF-LTD modifies Cs waveforms and attenuates CF-evoked dendritic calcium transients (*Weber et al., 2003*). In our study, optogenetic CF suppression likely simulated the functional consequences of CF-LTD, which may have impaired PF-PC LTD and subsequently hindered gain enhancement during learning (*Han et al., 2007*).

Our findings underscore a phase-specific role for CF input, demonstrating that CF activity is critical for motor memory acquisition but less necessary for consolidation and retrieval. Using optogenetic inhibition, we achieved precise temporal control over CF activity within distinct learning phases, enabling targeted manipulation of CF signaling. This differs from prior studies that employed pharmacological or lesion methods, which affect broader cerebellar pathways and may implicate CF signaling across multiple memory stages. The task demands inherent to different learning phases may explain these observed differences. Tasks requiring high-error correction, such as those encountered during the active learning phase, likely depend on CF-driven signals for precise error correction during acquisition. In contrast, CF-independent mechanisms may dominate later phases where such adjustments are less critical. This context-sensitive role highlights the adaptability of CF signaling in tuning cerebellar circuits to meet the specific demands of each learning phase. Moreover, cerebellar-dependent memory consolidation exhibits biphasic temporal dynamics: an initial rapid phase of consolidation followed by prolonged stabilization. Our study specifically focused on the early consolidation phase, emphasizing cerebellar plasticity mechanisms that are critical for encoding and stabilizing motor memory (*Cooke et al., 2004*; *Seo et al., 2024*; *Steinmetz and Freeman, 2016*; *Titley et al., 2007*). Importantly, changes during this early period significantly shape memory formation and retention, while alterations beyond this timeframe have minimal impact on storage or performance. While our findings align with this framework, extended consolidation periods may involve slower processes, such as systems-level integration into downstream nuclei or cortical regions, contributing to memory stabilization or generalization. Future investigations using chronic in-vivo recordings or live imaging could elucidate these prolonged dynamics and clarify CF transmission's nuanced contributions across phases of cerebellar-dependent memory consolidation.

In conclusion, inhibiting CF transmission during motor learning impairs gain increases specifically during acquisition. These findings provide critical insights into CF signaling's role in cerebellum-dependent learning while underscoring the need for further exploration of how cerebellar circuits contribute to adaptive motor memory over extended timeframes.

# Materials and methods

**Key resources table**

| Reagent type (species) or resource | Designation | Source or reference | Identifiers | Additional information |
|---|---|---|---|---|
| Other | AAV9.CaMKIIα.EGFP | Addgene | RRID:Addgene_50469 | Plasmid #50469 |
| Other | AAV1. CaMKIIα.eNpHR 3.0.EYFP | Addgene | RRID:Addgene_26971 | Plasmid #26971 |
| Chemical compound, drug | Zoletil | Virvac | | |
| Chemical compound, drug | Rompun | Bayer | | |
| Chemical compound, drug | Dexamethasone | Samyang Pharmaceutical | | |
| Chemical compound, drug | Meloxicam | Boehringer Ingelheim | | |
| Software, algorithm | ZEN | Zeiss | RRID:SCR_013672 | |
| Software, algorithm | Fiji | http://fiji.sc | RRID:SCR_002285 | |
| Software, algorithm | Patchmaster | HEKA | RRID:SCR_000034 | |
| Software, algorithm | Igor Pro | WaveMetrics | RRID:SCR_000325 | |
| Software, algorithm | GraphPad Prism | GraphPad | RRID:SCR_002798 | |

## Animals and stereotaxic surgery for virus injection

All experimental procedures were approved by the Seoul National University Institutional Animal Care and Use Committee (SNU-21031907–1 and SNU-230109-2-4). Animals were housed under a 12 hr light–dark schedule, with food and water available ad libitum, at 20–24 °C and 40–60% relative humidity. First, only male wild-type mice (aged 7–10 weeks C57Bl/6 N from Orient Bio) were anesthetized using intraperitoneal injections of a Zoletil/Rompun mixture (30 mg / 10 mg/kg). For optogenetic expression in CF, the virus was injected into the IO 2–3 weeks before the behavioral test, as previously described (*Kimpo et al., 2014*; *Roh et al., 2020*). Briefly, bilateral injections were made at the midpoint between the edge of the occipital bone and the C1 cervical vertebra. The glass pipette was set at a 55° angle from the vertical and 7° from the midline. After approaching a 3 mm depth, a virus solution containing 100–200 nL of AAV1-CaMkIIα-eNpHR3.0-EYFP or AAV9.CaMkIIα-EGFP was injected with a Picopump at 5 nL/s. The pipettes were left in place for 10 min before they were removed to minimize backflow.

## Stereotaxic surgery for optic cannula implantation

After the mice were fully recovered (a few days to a week) from the first surgery of virus injection, a second operation of optic cannula implantation (Ø1.25 mm Ceramic Ferrule, L=3.0 mm) near cerebellar flocculus and head fixation was conducted. Unlike the fiber optic cannula, head fixation parts were hand-made. Before the surgery, mice were anesthetized by intraperitoneal injection of Zoletil 50 (Virbac, 15 mg/kg) and xylazine (Rompun, Bayer, 15 mg/kg) mixture. Mice were given 48 hr to recover after surgery.

## Confirmation of viral expression

After completion of all the experiments, sampling using cardiac perfusion is performed. The brain was then extracted and coronal sections were made at 30 μm intervals. Images were acquired and processed using a confocal microscope (Zeiss LSM 7 MP, Carl Zeiss, Jena, Germany) and Zen software (Zeiss). The location of virus expression was confirmed by comparison with the brain atlas (Paxinos and Franklin's The Mouse Brain in Stereotaxic Coordinates 4th Edition by George Paxinos).

## Optical suppression of CF terminals in the cerebellar cortex in vivo

To manipulate CF transmission-induced Cs in PCs, AAV1.CaMKIIα.eNpHR3.0.eYFP or AAV8.CaMKIIa.eGFP was bilaterally injected into the IO. A glass micro-pipette was used to inject 200 nL per site. To optically suppress the CF terminals and record CF-PC synaptic response, a small cranial window was made above the cerebellar cortex lobule IV/V a week after the viral injection. After 2 weeks from the viral injection, mice underwent single-unit recordings. The recordings were performed as described

in Neural data acquisition and analysis. Cs responses were acquired during 10 s of yellow on and off sessions 20 times to test the effect of CF terminal suppression on CF-PC synaptic transmission.

## Neural data acquisition and analysis

Single-unit recordings were performed while mice were head-fixed and awake in the recording rack. A Digital Lynx system (Neuralynx, Bozeman, MT) was used to amplify, band pass filter (0.1–8000 Hz for simple spikes, 10–200 Hz for complex spikes), and digitize the electrode recordings. A silicon neural probe (Cambridge NeuroTech) was used to probe and acquire Cs signals. Cs signals were collected at a 32 kHz sampling rate. Cs waveforms were manually selected based on visual inspection of the averaged appearance of characteristic negative $Ca^{2+}$ peaks.

## Slice preparation, electrophysiology, and optogenetic validation

An acute brain slice preparation and electrophysiological experiments were carried out as previously described (*Lee et al., 2024*). First, mice aged 9–10 weeks with AAV1.CaMKIIα.eNpHR 3.0.WPRE-EYFP injection into their inferior olivary neuron 2–3 weeks before the preparation were anesthetized by isoflurane and briefly decapitated. Then, 250 μm-thick sagittal slices of the cerebellar vermis were obtained from the mice using a vibratome (VT1200S, Leica). The ice-cold cutting solution contained 75 mM sucrose, 75 mM NaCl, 2.5 mM KCl, 7 mM MgCl2, 0.5 mM CaCl2, 1.25 mM NaH2PO4, 26 mM NaHCO3, and 25 mM glucose with bubbled 95% O2 and 5% CO2. The slices were immediately transferred into the artificial cerebrospinal fluid (ACSF) containing 125 mM NaCl, 2.5 mM KCl, 1 mM MgCl2, 2 mM CaCl2, 1.25 mM NaH2PO4, 26 mM NaHCO3, and 10 mM glucose with bubbled 95% $O_2$ and 5% $CO_2$. They were allowed to recover at 32 °C for 30 min and at room temperature for 1 hr. All recordings were performed within 8 hr of recovery.

Brain slices were placed in a submerged chamber with perfusion of ACSF for at least 10 min before recording. We used OptoPatcher (HEKA Elektronik) holding recording pipette (2–3 MΩ) filled with internal solution containing 9 mM KCl, 10 mM KOH, 120 mM K-gluconate, 3.48 mM MgCl2, 10 mM HEPES, 4 mM NaCl, 4 mM Na2ATP, 0.4 mM Na3GTP, and 17.5 mM sucrose (pH 7.25). The stimulation electrode was placed on the granule cell layer near the PC. The stimulation isolator injected a brief current pulse to the stimulation electrode and was controlled using PatchMaster software (HEKA Elektronik). Stimulation intensity (6–30 μA) was calibrated to minimally evoke CF EPSC. Regarding the laser on trial, a 593.5 nm irradiation from a yellow laser was applied 5 s before CF stimulation. The eEPSC amplitude was averaged over three lasers on and off trials. Under current clamp mode, complex spikes induced by CF stimulation were recorded using the same validation protocol. Overall, the laser illumination (3–5 mW) consistently inhibited CF transmission. Electrophysiological data were acquired using an EPC10 patch-clamp amplifier (HEKA Elektronik) and PatchMaster software (HEKA Elektronik) with a sampling frequency of 20 kHz, and the signals were filtered at 2 kHz. All electrophysiological recordings were acquired from the central cerebellar vermis. The amplitude of EPSC was analyzed using Igor Pro (WaveMetrics).

## Optokinetic reflex behavior test

First, two sessions of acclimation were conducted. During each session, the mouse was restrained using a custom-made restrainer for 20 min both with and without light for habituation to the recording environment. Following the procedure reported by *Stahl et al., 2000*, a calibration was performed. This allowed for the conversion of the dynamics of pupil-to-eye rotation. Next, three basal oculomotor performances, including (OKR), vestibulo-ocular reflex in the dark (dVOR), and vestibulo-ocular reflex in the light (lVOR), were measured. During the recording of eye movements, it was necessary to control pupil dilation that was induced by an absence of light in dark conditions. Thus, physostigmine salicylate solution (Eserine; Sigma Millipore) was given to mice under brief isoflurane anesthesia. The concentration of eserine solution was increased from 0.1%, 0.15%, and 0.2% based on the pupil size. Basal oculomotor performances were described in gains. For the actual learning protocol, we adopted OKR, and it consisted of five training sessions (10 min per session), six checkup points, and 24 hr of consolidation period. During the training sessions, the mice were visually stimulated with a sinusoidally rotating drum (±5 degrees). Completely trained mice were placed back into their home cage, which was stored in the dark condition until the last gain checkup points.

## Experimental design

For our in vivo OKR experiments, group sizes for mice expressing AAV-Ef1α-DIO-NpHR-EYFP (NpHR) or AAV-Ef1α-DIO-EYFP (EYFP) were chosen to match those of comparable published studies (PMID: 36639897; 16458438; 16135754; 11805298) to ensure adequate statistical power. Each cohort underwent identical surgical, recovery, and behavioral testing pipelines, with two independent cohorts run to confirm reproducibility. Mice were randomly assigned to the NpHR or EYFP groups, and the order of OKR testing was shuffled across animals to eliminate potential time-of-day or operator effects. While full blinding to viral condition was not technically feasible, all data acquisition and processing steps were standardized to minimize bias.

## Gain analysis

Gains acquired from basal oculomotor performances and OKR learning were calculated as the ratio of evoked eye movements to the movement of the screen or turn table as visual or vestibular stimuli, respectively. A custom-built LabView (National Instruments) analysis tool was used for all the calculations.

## Statistical analysis

Data are presented as mean ± SEM and were analyzed in GraphPad Prism (GraphPad Software Inc, CA, USA). Between-group comparisons of continuous variables (e.g., CF-EPSC suppression magnitudes, percent gain change, consolidation percentages) were analyzed by unpaired two-tailed t-tests. Within-animal comparisons (e.g. pre- versus post-learning gain; number of complex-spike spikelets) were analyzed by paired two-tailed t-tests. Simple and complex spike firing rates were compared with two-tailed Mann-Whitney U tests, which did not meet parametric assumptions and/or involved small sample sizes. All tests were two-tailed, and statistical significance was defined as $p < 0.05$. In figures, asterisks denote significance levels (*$p < 0.05$; **$p < 0.01$; ***$p < 0.001$; ****$p < 0.0001$), and the exact test used with sample sizes and p-values is reported in each legend.

## Acknowledgements

This work was supported by National Research Foundation of Korea (NRF) grants funded by the Korean Government (MSIT) (NRF-2018R1A5A2025964 and 2022M3E5E8017970 to Sang Jeong Kim).

## Additional information

### Funding

| Funder | Grant reference number | Author |
|---|---|---|
| National Research Foundation of Korea | 2018R1A5A2025964 | Sang Jeong Kim |
| National Research Foundation of Korea | 2022M3E5E8017970 | Sang Jeong Kim |
| National Research Foundation of Korea | RS-2025-00553878 | Jaegeon Lee |

The funders had no role in study design, data collection and interpretation, or the decision to submit the work for publication.

### Author contributions

Jewoo Seo, Conceptualization, Data curation, Formal analysis, Validation, Investigation, Methodology, Writing - original draft, Writing – review and editing; Seung Ha Kim, Data curation, Formal analysis, Validation, Investigation, Methodology, Writing – review and editing; Jaegeon Lee, Formal analysis, Validation, Investigation, Methodology, Writing – review and editing; Min Seok Kim, Data curation, Formal analysis, Validation, Investigation, Methodology; Yong-Seok Lee, Supervision, Project administration, Writing – review and editing; Sang Jeong Kim, Resources, Supervision, Funding acquisition, Project administration, Writing – review and editing

## Author ORCIDs
Jewoo Seo https://orcid.org/0000-0002-9162-848X
Seung Ha Kim https://orcid.org/0000-0003-0279-8290
Jaegeon Lee https://orcid.org/0000-0001-7865-3966
Min Seok Kim https://orcid.org/0009-0002-3478-7975
Yong-Seok Lee https://orcid.org/0000-0002-6217-9574
Sang Jeong Kim https://orcid.org/0000-0001-8931-3713

## Ethics
The Seoul National University Institutional Animal Care and Use Committee approved all experimental procedures. All of the animals were handled according to the approved institutional animal care and use committee (IACUC) and Institutional Biosafety Committee (IBC) of the Seoul National University College of Medicine (SNU-21031907-1 and SNU-230109-2-4).

Reviewer #2 (Public review): https://doi.org/10.7554/eLife.95838.4.sa1
Reviewer #3 (Public review): https://doi.org/10.7554/eLife.95838.4.sa2
Author response https://doi.org/10.7554/eLife.95838.4.sa3

---

# Additional files

### Supplementary files
MDAR checklist

### Data availability
All data generated or analysed during this study are included in the manuscript and supporting files; source data files have been provided for Figures 1–4.

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
