## [Editor Report · eLife Assessment]

This study presents potentially **valuable** insights into the role of climbing fibers in cerebellar learning. The main claim is that climbing fiber activity is necessary for optokinetic reflex adaptation, but is dispensable for its long-term consolidation. There is evidence to support the first part of this claim, though it requires a clearer demonstration of the penetrance and selectivity of the manipulation. However, support for the latter part of the claim is **incomplete** owing to methodological concerns, including the robustness of the CF marking and manipulation approach and the unclear efficacy of longer-duration climbing fiber activity suppression.

---

## [Referee Report · Reviewer #2 (Public review)]

Summary:

The authors aimed to explore the role of climbing fibers (CFs) in cerebellar learning, with a focus on optokinetic reflex (OKR) adaptation. Their goal was to understand how CF activity influences memory acquisition, memory consolidation, and memory retrieval by optogenetically suppressing CF inputs at various stages of the learning process.

Strengths:

The study addresses a significant question in the cerebellar field by focusing on the specific role of CFs in adaptive learning. The authors use optogenetic tools to manipulate CF activity. This provides a direct method to test the causal relationship between CF activity and learning outcomes.

Weaknesses:

Despite shedding light on the potential role of CFs in cerebellar learning, the study is hampered by significant methodological issues that question the validity of its conclusions. The absence of detailed evidence on the effectiveness of CF suppression and concerns over tissue damage from optogenetic stimulation weaken the argument that CFs are not essential for memory consolidation. These challenges make it difficult to confirm whether the study's objectives were fully met or if the findings conclusively support the authors' claims. The research commendably attempts to unravel the temporal involvement of CFs in learning but also underscores the difficulties in pinpointing specific neural mechanisms that underlie the phases of learning. Addressing these methodological issues, investigating other signals that might instruct consolidation, and understanding CFs' broader impact on various learning behaviors are crucial steps for future studies.

Comments on revisions:

In this revision, the authors provide new data regarding the effect of eNpHR on CF-evoked complex spiking in vivo but fails to address overall concerns showing the functional effect that explains their causal results. Additionally, the paper has a narrow "CF-or-nothing" framing that leaves unanswered the central question of which signal instructs consolidation if CFs do not. Substantial new experiments and tighter logic are required before the work can serve as a definitive test of CF involvement in different memory processes.

---

## [Referee Report · Reviewer #3 (Public review)]

Summary:

The authors attempted to study connections with the inferior olive to the cerebellar cortex and analyze impacts on optokinetic reflex using optogenetics to perturb the pathway. This is a commendable effort as these methods are very challenging due to the location of the inferior olive and recording methods.

Strengths:

The authors have shown that climbing fiber activity was altered due to the optogenetic perturbation. They have added an additional figure to show that complex spikes disappear with inhibitory optogenetics and the impacts on behavior are interesting.

Weaknesses:

The images provided to show injection region are difficult to see and specific cell types are not co-labeled. The data and strength of the results would benefit from high-resolution images demonstrating selectivity and expression, in particular for Figure 2A and 3A. In addition, while the processed recording data looks very striking, including the raw data, as done in Figure 2, would again support the conclusions.

One major concern is that the viruses chosen are non-specific to the cell targets and a cre-based approach is lacking to draw conclusions on only the targeted pathway of interest. It is unclear based on the figures provided if the AAVs labeled only the pathway of interest. It would be interesting to know if typical memory acquisition returns in the same animals if inhibition stops and if animal movement was impacted by the perturbation.

---

## [Author Response]

The following is the authors’ response to the previous reviews

**Public Reviews:**

**Reviewer #1 (Public Review):**
Summary:The study by Seo et al highlights knowledge gaps regarding the role of cerebellar complex spike (CS) activity during different phases of learning related to optokinetic reflex (OKR) in mice. The novelty of the approach is twofold: first, specifically perturbing the activity of climbing fibers (CFs) in the flocculus (as opposed to disrupting communication between the inferior olive (IO) and its cerebellar targets globally); and second, examining whether disruption of the CS activity during the putative "consolidation phase" following training affects OKR performance.The first part of the results provides adequate evidence supporting the notion that optogenetic disruption of normal CF-Purkinje neuron (PN) signaling results in the degradation of OKR performance. As no effects are seen in OKR performance in animals subjected to optogenetic irradiation during the memory consolidation or retrieval phases, the authors conclude that CF function is not essential beyond memory acquisition. However, the manuscript does not provide a sufficiently solid demonstration that their long-term activity manipulation of CF activity is effective, thus undermining the confidence of the conclusions.Strengths:The main strength of the work is the aim to examine the specific involvement of the CF activity in the flocculus during distinct phases of learning. This is a challenging goal, due to the technical challenges related to the anatomical location of the flocculus as well as the IO. These obstacles are counterbalanced by the use of a well-established and easy-to-analyse behavioral model (OKR), that can lead to fundamental insights regarding the long-term cerebellar learning process.Weaknesses:The impact of the work is diminished by several methodological shortcomings.Most importantly, the key finding that prolonged optogenetic inhibition of CFs (for 30 min to 6 hours after the training period) must be complemented by the demonstration that the manipulation maintains its efficacy. In its current form, the authors only show inhibition by short-term optogenetic irradiation in the context of electrical-stimulation-evoked CSs in an ex vivo preparation. As the inhibitory effect of even the eNpHR3.0 is greatly diminished during seconds-long stimulations (especially when using the yellow laser as is done in this work see Zhang, Chuanqiang, et al. "Optimized photo-stimulation of halorhodopsin for long-term neuronal inhibition." BMC biology 17.1 (2019): 1-17), we remain skeptical of the extent of inhibition during the long manipulations. In short, without a demonstration of effective inhibition throughout the putative consolidation phase (for example by showing a significant decrease in CS frequency throughout the irradiation period), the main claim of the manuscript of phase-specific involvement of CF activity in OKR learning can not be considered to be based on evidence.Second, the choice of viral targeting strategy leaves gaps in the argument for CF-specific mechanisms. CaMKII promoters are not selective for the IO neurons, and even the most precise viral injections always lead to the transfection of neurons in the surrounding brainstem, many of which project to the cerebellar cortex in the form of mossy fibers (MF). Figure 1Bii shows sparsely-labelled CFs in the flocculus, but possibly also MFs. While obtaining homogenous and strong labeling in all floccular CFs might be impossible, at the very least the authors should demonstrate that their optogenetic manipulation does not affect simple spiking in PNs.Finally, while the paper explicitly focuses on the effects of CF-evoked complex spikes in the PNs and not, for example, on those mediated by molecular layer interneurons or via direct interaction of the CF with vestibular nuclear neurons, it would be best if these other dimensions of CF involvement in cerebellar learning were candidly discussed.
**Reviewer #2 (Public Review):**
Summary:The authors aimed to explore the role of climbing fibers (CFs) in cerebellar learning, with a focus on optokinetic reflex (OKR) adaptation. Their goal was to understand how CF activity influences memory acquisition, memory consolidation, and memory retrieval by optogenetically suppressing CF inputs at various stages of the learning process.Strengths:The study addresses a significant question in the cerebellar field by focusing on the specific role of CFs in adaptive learning. The authors use optogenetic tools to manipulate CF activity. This provides a direct method to test the causal relationship between CF activity and learning outcomes.Weaknesses:Despite shedding light on the potential role of CFs in cerebellar learning, the study is hampered by significant methodological issues that question the validity of its conclusions. The absence of detailed evidence on the effectiveness of CF suppression and concerns over tissue damage from optogenetic stimulation weakens the argument that CFs are not essential for memory consolidation. These challenges make it difficult to confirm whether the study's objectives were fully met or if the findings conclusively support the authors' claims. The research commendably attempts to unravel the temporal involvement of CFs in learning but also underscores the difficulties in pinpointing specific neural mechanisms that underlie the phases of learning. Addressing these methodological issues, investigating other signals that might instruct consolidation, and understanding CFs' broader impact on various learning behaviors are crucial steps for future studies.

We appreciate the editors and reviewers for their constructive feedback and careful consideration of our manuscript. Despite their acknowledgment of the potential of our study to yield valuable insights into the role of CF activity in cerebellar learning and its phase-specific involvement, we have meticulously addressed all the methodological concerns raised by providing additional clarifications and explanations in this letter.

In response to concerns regarding the efficacy of long-term optogenetic inhibition, we conducted additional in vivo monitoring of CF activity during the irradiation period, confirming sustained inhibition of complex spikes throughout the consolidation phase (Figure 2, lines 112-139). Although stable single-unit recording beyond 40 minutes was not feasible due to technical challenges, the robust suppression of CF-evoked complex spikes we observed during this period (Figure 2, lines 112–139) provides strong evidence that halorhodopsin-mediated inhibition persists over the longer irradiation intervals employed in our behavioral assays.

Moreover, given that there is a concern regarding the CaMKII promoter also inducing expression in neighboring mossy fibers, potentially affecting simple spike activity, we have presented data in Figure 2C, which illustrates that PC simple spike firing rates remain unchanged during prolonged illumination. This finding confirms that our optogenetic manipulation selectively disrupts CF-mediated complex spikes without influencing mossy fiber to PC transmission. We have elucidated these results further in lines 128 to 136.

Lastly, we have broadened our Discussion to consider alternative mechanisms of CF involvement in cerebellar learning, including the modulation of molecular layer interneurons (Rowan et al., 2018) and direct CF interactions with vestibular nuclear neurons (Balaban et al., 1981), thereby offering a more comprehensive perspective on the multifaceted role of CF signaling. Specific clarifications regarding these points are articulated from lines 222 to 242 and 243 to 254 in the manuscript. We are confident that these revisions adequately address the reviewers' concerns and further substantiate the specificity and significance of our study findings

(1) Rowan, Matthew JM, et al. "Graded control of climbing-fiber-mediated plasticity and learning by inhibition in the cerebellum." *Neuron* 99.5 (2018): 999-1015.

(2) Balaban, Carey D., Yasuo Kawaguchi, and Eiju Watanabe. "Evidence of a collateralized climbing fiber projection from the inferior olive to the flocculus and vestibular nuclei in rabbits." *Neuroscience letters* 22.1 (1981): 23-29.